# Analysis and Forecast of Indicators Related to Medical Workers and Medical Technology in Selected Countries of Eastern Europe and Balkan

**DOI:** 10.3390/healthcare11050655

**Published:** 2023-02-23

**Authors:** Milos Stepovic, Stefan Vekic, Radisa Vojinovic, Kristijan Jovanovic, Snezana Radovanovic, Svetlana Radevic, Nemanja Rancic

**Affiliations:** 1Faculty of Medical Sciences, University of Kragujevac, 34000 Kragujevac, Serbia; 2Faculty of Economics, University of Belgrade, 11000 Belgrade, Serbia; 3Department of Radiology, Faculty of Medical Sciences, University of Kragujevac, 34000 Kragujevac, Serbia; 4Department of Anatomy, Faculty of Medical Sciences, University of Kragujevac, 34000 Kragujevac, Serbia; 5Department of Social Medicine, Faculty of Medical Sciences, University of Kragujevac, 34000 Kragujevac, Serbia; 6Medical Faculty of the Military Medical Academy, University of Defence in Belgrade, 11000 Belgrade, Serbia; 7Centre for Clinical Pharmacology, Military Medical Academy, 11000 Belgrade, Serbia

**Keywords:** health indicators, medical workers, medical technology, Eastern Europe, Balkan

## Abstract

Health indicators measure certain health characteristics in a specific population or country and can help navigate the health systems. As the global population is rising, the demand for an increase in the number of health workers is simultaneously rising. The aim of this study was to compare and predict the indicators related to the number of medical workers and medical technologies in selected countries in Eastern Europe and Balkan in the studied period. The article analyzed the reported data of selected health indicators extracted from the European Health for All database. The indicators of interest were the number of physicians, pharmacists, general practitioners and dentists per 100,000 people. To observe the changes in these indicators through the available years, we used linear trends, regression analysis and forecasting to the year 2025. The regression analysis shows that the majority of the observed countries will experience an increase in the number of general practitioners, pharmacists, health workers/professionals and dentists, as well as in the number of computerized tomography scanners and the number of magnetic resonance units, predicted to occur by 2025. Following trends of medical indicators can help the government and health sector to focus and navigate the best investments for each country according to the level of their development.

## 1. Introduction

Health indicators measure certain health characteristics in a specific population or country [1]. Health indicators aim to describe and monitor the health status of the population. There are many definitions of health indicator which are designed by important institutions and organizations. Some of the reasons for the utilization of health indicators are program management, resource allocation, country progress monitoring, performance-based payment and global reporting [2]. Health indicators can be sorted into four different spheres: indicators of health status, indicators of the health system, indicators of health status and indicators of service coverage. Information concerning health workers, health financing and quality of healthcare and medical information can be provided from these indicators [3,4].

Throughout the 20th century, health systems have made a huge contribution to better health in the majority of the world’s population [5]. Health systems currently play a larger and more influential role in people’s lives than in the past. During the last century, health systems were subjected to different reforms, such as the establishment of different healthcare systems and propagation of social security. Primary healthcare became a path towards the ultimate goal of every health system, which is universal health coverage, affordable to all [6,7]. The goal is not only to achieve care for all but also provide all with quality basic care, defined mainly by the criteria of efficiency, cost and social acceptability [8,9].

As the global population is rising, the demand for the increase in the number of health workers is simultaneously rising. The United Nations High-Level Commission on Health Employment and economic growth projected that compared to the 2013 population, 80 million health workers will be needed just to keep up with the demands of the global population, compared to the number of health workers worldwide at the moment of projection, which leaves a gap of 18 million [10]. Such a large gap will influence the likelihood of the global community achieving universal health coverage. The lack of medical workers will also influence the quality of healthcare provided for the less-developed parts of countries and will mostly affect rural areas [11]. Inadequate health availability will end with a rise in communicable and non-communicable diseases, becoming a large part of the global burden of disease; it will create a higher necessity for additional tests, drugs and different and expensive technologies that may be lacking with respect to the numbers per capita worldwide, and eventually, it will increase mortality and morbidity rates. Another problem is the migration of health workers, thus leaving some of the less-developed countries with even bigger problems [12]. Citizens older than 65 years are especially vulnerable as their proportion is rising every year. With this global aging also comes health expenditures for different drugs, treatments and necessary tests and examinations due to the likely presence of comorbid diseases. Countries in different states of development will be able to invest varying amounts of GDP in healthcare, which will create further problems. Communicable diseases are also unable to be entirely removed as they are also present in, and not typical for, any age group. During the COVID-19 pandemic, the necessity for good health system organization and sufficient medical workers and technologies was imperative [13].

The last few decades have demonstrated that medical technology utilization accounts for the greatly increased percentage of health spending (of nearly 50%). New technologies improve medical care, but as mentioned they also influence the rise in healthcare expenditures that affects both governmental and individual budgets [14].

Eastern Europe and Balkan, the Russian Federation and the former Union of Soviet Social Republics, and Turkey are countries that have a shared historical background; therefore, the way they manage different economic crises affects their similarly organized health systems. These countries have very diverse population structures when considering religion (Catholicism, Orthodox Christianity and Islam), which is very important when considering their historical approaches towards creating important solutions compared to Northern and Southern Europe. When developing their health systems, these countries also had very dependent relationships between each other, as the health systems that these countries used were those that others adapted to their own countries. There were three systems of health financing that were dominant throughout the 19th century: the Bismarck, Beveridge, and Semashko systems [15]. Progress in medical technology and pharmaceuticals were hard to follow, as the other, more developed countries of Europe and the countries that we selected had large problems in accessing medical healthcare, especially in the rural and less-developed parts of countries. A great deal of medical expenditure was paid out-of-pocket. Those countries were less industrially developed at the time, especially during the Cold War, and much of their GDP came from their agricultural economy, which could not endure such a fast medical development, particularly when population aging became more prominent. The socioeconomic situation of these countries was weak, so healthcare was quite expensive and less available to all [16].

The aim of this study was to compare and predict the indicators related to the number of medical workers and medical technologies in selected countries of Eastern Europe and Balkan with similar historical backgrounds in the development of their health systems. The problem of the insufficient number of medical workers and technologies in some of these countries, due to different scenarios, can affect the health coverage of citizens and health organizations. Similar articles can help notice and prevent these problems. 

## 2. Materials and Methods

This study was conducted as a descriptive data analysis of observed indicators of interest—indicators of medical workers and indicators of health expenditures. The data source was the European Health for All database (HFA-DB), where Member States of the WHO (Geneva, Switzerland) European Region have been reporting essential health-related statistics since the 1980s [17]. This database is a cluster of different indicators that are part of major monitoring frames; it is based on reports, not estimates, and provides a large range of following years.

The selected indicators were physicians per 100,000 inhabitants, pharmacists (PP) per 100,000 inhabitants, general practitioners (PP) per 100,000 inhabitants and practicing dentists per 100,000 inhabitants. PP is an abbreviation for practicing physician/persons. All medical indicators used in this article provide some aspect of medical health; persons in the educational process were not included in these indicators. Every used indicator has inclusion and exclusion criteria defined by the World Health Organization before being inputted into the HFA-DB. Indicators of increased medical expenditures were also analyzed: the total number of computer tomography scanners per 100,000 inhabitants and the total number of magnetic resonance imaging units per 100,000 inhabitants. The included countries were: Albania, Bulgaria, Bosnia and Herzegovina, Belarus, Greece, Croatia, North Macedonia, Montenegro, Romania, the Russian Federation, Serbia, Slovenia, Turkey, Estonia, Lithuania, Latvia and Ukraine. The observation period was from 1990 to 2016 (the last year available from HFA-DB after the database update in September 2022). Countries without consistent following of the defined indicators were not included in the analysis. Years that were observed varied between the countries; therefore, the first year used was the year that most of the countries had in common, and the last year used was 2014 or 2016.

As we were observing changes in these indicators only through time (continuous variable), a linear trend was chosen for the analysis [18]. With this data, we were able to access the current simple linear trends using the Excel mathematics algorithm and construct the graphs that showed us the changes in those trends. Linear regression predicted values based on the data from the available two and a half decades [19,20]. Forecasting techniques are commonly utilized for historical data, as is the case in our research, and we used medium-term forecasting analysis, anticipating several years in advance (to the year 2025). Forecasting analysis was performed by combining Excel analysis and IBM SPSS program version 26.0. SPSS is an IBM (Armonk, NY, USA) product designed for statistical analysis, predictive analysis, big data integration and similar algorithms. Only one decade after the last available year was predicted with the purpose of tracking the current trends. A regression line uses a formula to calculate its predictions: Y = A + BX. Y is the dependent variable, X is the independent variable, B is the slope of the line and A is the point where Y intercepts the line. Regression gives an R-squared value; the values range from 0 to 1, with 0 being a weaker model and 1 being a stronger model. The confidence interval for prediction was 95%. Interquartile range 25–75th percentile was calculated with the purpose of enhancing the accuracy of dataset statistics by dropping lower contributions. The median operation was calculated for each country and indicator for easier comparison.

The data were anonymous and do not belong to individual citizens. According to the International Ethical Guidelines for Biomedical Research involving Humans and Good Clinical Practice Guidelines, a study like this does not require consideration by the Ethics Committee, as per the International Ethical Guidelines for Health-related Research Involving Humans (https://cioms.ch/wp-content/uploads/2017/01/WEB-CIOMS-EthicalGuidelines.pdf, accessed on 9 January 2023) and European Medicine Agency (Amsterdam, The Netherlands) (https://www.ema.europa.eu/en/ich-e6-r2-good-clinical-practice-scientific-guideline, accessed on 9 January 2023).

## 3. Results

### 3.1. Number of Medical Workers per 100,000 Inhabitants

The number of general practitioners per 100,000 inhabitants had the highest median values in Latvia (71.5) and Serbia (71.4), while the lowest median value was in Belarus (8.7) (Figure 1a). The regression analysis shows that in all observed countries there was an increase in this number, with the highest in Latvia (y = 2.4167x + 52.227; R^2^ = 0.95) and Ukraine; only in Albania (y = −0.9346x + 56.908; R^2^ = 0.0911) did we observe a decrease. North Macedonia and Lithuania did not have data on this indicator (Table 1). The number of general practitioners per 100,000 inhabitants is expected to increase by 2025, compared to the last available year, in 13 observed countries, the highest in Latvia by approximately 26; while in two countries, Romania and Bulgaria, a decrease in this number can be expected, by approximately 6 and 4.5 fewer, respectively, compared to the last available year.

The number of pharmacists per 100,000 inhabitants had the highest median values in Greece (96/100,000) and Lithuania (63/100,000), while the lowest median values were in Ukraine (3) and Russia (Figure 1b). The regression analysis shows that in 15 of the 17 observed countries, there was an increase in the number of pharmacists per 100,000 inhabitants; the highest in Croatia (y = 1.6326x + 31.777; R^2^ = 0.985) and Slovenia. The decline in the number of pharmacists was particularly observed in Bulgaria (y = −1.5415x + 28.358; R^2^ = 0.7519) and in Latvia (Table 2). The number of pharmacists per 100,000 inhabitants is expected to increase by 2025 in 13 observed countries, compared to the last observed year, and the largest increases can be expected in Romania and Greece. A decrease is expected in three countries, and the largest decreases are expected in Bulgaria and Albania.

The number of health workers per 100,000 inhabitants had the highest median values in Greece (466) and Latvia (372), while the lowest median values were in Albania (128) and Turkey (Figure 1c). The regression analysis shows that in 15 of the 17 observed countries, an increase in the number of health workers per 100,000 inhabitants occurred, and the highest increases were in Turkey (y = 3.5474x + 94.603; R^2^ = 0.9923) and Croatia. A decrease in the number of health workers was observed in Albania (y = −3.0557x + 152.14; R^2^ = 0.2289) and in Macedonia (Table 3). The number of medical workers per 100,000 inhabitants by 2025 is expected to increase in 15 observed countries, with the most in Greece by 189 compared to the last observed year. A decline can be expected in Russia, by 50% less compared to the last observed year, as well as in Albania.

The highest median value of the indicator number of dentists working per 100,000 inhabitants in the observed period was observed in Bulgaria, with 82 per 100,000 inhabitants, as well as in Estonia, while the lowest median values were observed in Bosnia and Herzegovina, with 19 per 100,000 inhabitants (Figure 1d). Regression analysis shows that there is a growing trend in the indicator of the number of dentists working per 100,000 in most observed countries, with the most pronounced growth occurring in Slovenia (y = 2.5453x + 54.943; R^2^ = 0.9631) and Romania. A downward trend is expected in three countries, with the most pronounced in Albania (y = −2.5801x + 53.951; R^2^ = 0.9068) (Table 4). The number of dentists working per 100,000 inhabitants will increase by 2025 in almost all observed countries, and the highest will be in Ukraine, by 32 more than in 2013.

### 3.2. Number of Medical Technologies Used in Health Services

The highest median value of the total number of computerized tomography scanners per 100,000 inhabitants in the observed period was observed in Latvia and Bulgaria, with 3 per 100,000 inhabitants, while the lowest median value was observed in Romania, with 0.8 per 100,000 inhabitants (Figure 2a). Regression analysis shows that there is a growing trend in the indicator of the total number of scanners for computed tomography per 100,000 inhabitants in 9 out of 10 observed countries, with the most pronounced growth occurring in Romania (y = 0.1152x − 0.0636; R^2^ = 0.9945) and Latvia. A downward trend is expected only in Slovenia (y = −0.0112x + 1.1727; R^2^ = 0.1119) (Table 5). The indicator of the total number of scanners for computed tomography per 100,000 inhabitants will increase by 2025 in 9 out of 10 observed countries, with the most in Latvia, by 1.9 more than in 2016. It is expected that by 2025, this indicator will decrease only in Slovenia, by 0.2 less compared to 2016. 

The highest median value of the total number of magnetic resonance units per 100,000 inhabitants in the observed period was observed in Greece, with 2.2 per 100,000 inhabitants, and Turkey, while the lowest median values were observed in Serbia and Croatia, with 0.3 per 100,000 inhabitants (Figure 2b). Regression analysis shows that there is a growing trend in the indicator of the total number of magnetic resonance units per 100,000 inhabitants in 9 out of 10 observed countries, with the most pronounced growth occurring in Latvia (y = 0.1021x + 0.1864; R^2^ = 0.9743) and Romania. Only in Serbia is no change expected (Table 6). The indicator of the total number of magnetic resonance imaging units per 100,000 inhabitants will increase by 2025 in 9 of the 10 observed countries, and the highest in Lithuania, by 1 more than in 2016. It is expected that by 2025, this indicator will only remain the same in Serbia.

## 4. Discussion

Population aging is correlated with the rise of medical costs, which is becoming an established issue not only in Eastern European and Balkan countries but worldwide [21]. With numerous advances in technology, medicine and pharmaceuticals, people are living longer today than in previous decades, leading to increased healthcare costs [22,23]. Our research shows increasing numbers of medical workers in most of the observed countries, which are desirable nowadays as population aging and chronic diseases are also rising. A few countries show opposite results, which may indicate a problem forming in less-developed countries. This negative trend will continue if these countries do not undertake better organization of their health systems. 

Countries without established health systems that recognize the needs of elderly people will eventually suffer from large expenditure in national and out-of-pocket expenses [24]. Elderly people may suffer from two or more combined non-communicable diseases or undergo more surgical interventions, or laboratory analyses and various radiological imaging methods [25,26]. Radiographs and computed tomography have an important role in many aspects of diagnosis and evaluation of pathologies, and CBCT is used widely in dental practice with a reduced radiation dose compared to classical CT [27]. The usage and adaptation of new technologies is particularly challenging for many more developed countries as well, such as OEC countries. Research into the health economy recognizes CT and MRI as causes of increased medical expenditure, but on the other hand, they are also key technologies mostly used in different research in various fields of medicine and dentistry, so their accessibility may also indicate the better organization of health systems [28]. He et al. found that macroeconomic and socio-economic indicators have a significant correlation with the allocation of scans used in radiology and also with several health professionals [29]. Our research shows a growing trend over the observed time period in the number of CT scanners and MRI units in all observed countries, and our predictions show that this number per 100,000 inhabitants will continue to grow. Latvia and Bulgaria have the highest number of CT scanners and Greece has the highest number of MRI units.

The number of medical workers, doctors, pharmacists and dentists is also increasing in all observed countries in our research, which indicates an increased investment in health by the state. It is also predicted that this growth trend will continue until 2025. According to the 2018 predictions of The Department of Health, the percentage of the workforce in primary healthcare must rise by almost 50% by 2031 to meet the demands and reforms of health services [30]. Data about trends in those numbers are needed for the determination of the necessary capacity of health systems, because good planning without investigation is not possible [31]. Public health services had to adapt to the many challenges during the COVID-19 pandemic, especially due to a lack of medical workers. One of the reasons why some countries, particularly less-developed countries, had this issue, is due to outflow to the larger and more developed countries [32,33]. 

The number of dentists is expected to grow the most in Bulgaria, and most countries will increase their number of dentists per 100,000 inhabitants, with the exception of three countries according to the results of our research. The elderly population is a special group in the oral health sector because of their specific needs and therapy compared the younger people. The complexity of their dental therapy is additionally challenged by multiple co-morbidities [34]. As with the outflow of medical professionals, the mobility of dental doctors is raising problems in many countries [35]. Many studies have shown different factors that promote the mobility of medical workers in general; some of the main reasons are economy-related, such as searching for employment or higher salaries, but also, in the younger population, factors include higher education and improvement [36,37]. 

Pharmacists are considered the healthcare profession that is the most accessible, and the capacity of pharmacists is related to economic indicators, whereby the countries with weaker economic indicators have less workforce availability, which is directly correlated with inequalities faced by different socio-economic groups [38]. However, pharmacist workforce shortages have been reported in all sectors. It is highly recommended to follow the trends of this indicator globally for the future capacity of pharmacists. [39]. Our research shows that the lowest median value of the number of pharmacists per 100,000 inhabitants were found in Ukraine and Russia. Looking at the linear trend, most countries showed positive trends in the number of pharmacists, and prediction values to 2025 also follow these results, with the exception of three observed countries, most noticeable in Bulgaria and Albania.

Epidemiological transitions have a large impact on the health systems of countries, creating difficult challenges for healthcare providers [40]. The burden of disease is shifted through the transition (in earlier periods, infectious diseases dictated how much the state would spend on health; now this role is filled by chronic non-infectious diseases—diseases of well-being) and this is the reason why the health system must adapt and integrate new technologies [41,42]. A report about universal health coverage from the WHO presented large investments of nearly 10% of GDP on health, whereby average per capita spending is about USD 1000. [43]. The largest percentage of these costs were related to medicaments, treatment of inpatients and outpatients, tests, and scans, thus relegating the importance of prevention and preventive programs to the background [44].

## 5. Conclusions

Knowledge of changing trends in medical staff and medical technology is of crucial importance in the better re-composition of health sector needs. Universal health coverage is a main aim in the health sector of each country worldwide, which is hardly likely to succeed even with the best organization. Medical professionals are integral parts of every health organization, and with an insufficient number of workers, these aims would be even more unreachable. Additionally, new medical technology must follow the increasing trends and demands of the people in need of it. Government investments must follow the need for a higher number of medical workers. According to our research, there is mostly a positive trend in the number of medical workers and medical technology in the countries of Balkan and South-Eastern Europe, with a few exceptions where this trend is one of slow decrease. Following these indicators, the government and health sectors can focus on and navigate the best investments to influence the options for better health coverage in each country, with careful specification of the level of each country’s development. The importance of observing different health- and economic-related indicators is useful for different countries, and their analysis can be a reflection of the successful application of preventative measures. Assessing these trends and comparisons between countries may give valuable information about the organization of different health systems, and countries with a specific problem can adapt their health system accordingly.

## Figures and Tables

**Figure 1 healthcare-11-00655-f001:**
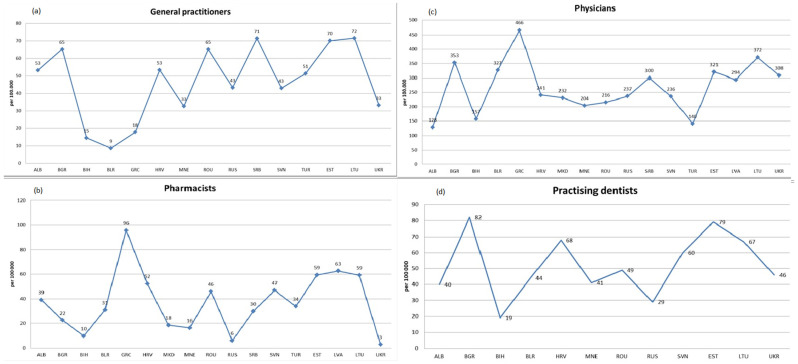
Median values of indicators: (**a**) **general practitioners**, (**b**) **pharmacists**, (**c**) **physicians**, (**d**) **practicing dentists**, per 100,000 inhabitants. Albania—ALB, Bosnia and Herzegovina—BIH, Bulgaria—BGR, Greece—GRC, Croatia—HRV, Montenegro—MNE, Northern Macedonia—MKD, Romania—ROU, Serbia—SRB, Slovenia—SVN, Turkey—TUR, Russia—RUS, Belarus—BLR, Lithuania—LTU, Latvia—LVA, Estonia—EST, Ukraine—UKR.

**Figure 2 healthcare-11-00655-f002:**
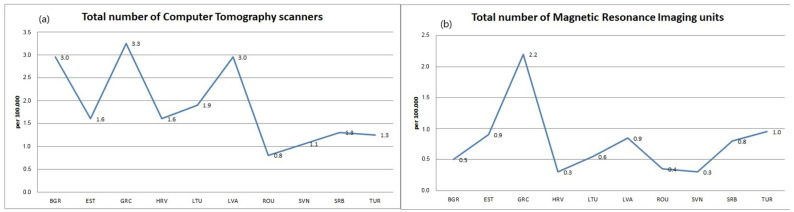
Median values of indicators: (**a**) **total number of computed tomography scanners** and (**b**) **magnetic resonance imaging units** per 100,000 inhabitants. Albania—ALB, Bosnia and Herzegovina—BIH, Bulgaria—BGR, Greece—GRC, Croatia—HRV, Montenegro—MNE, Northern Macedonia—MKD, Romania—ROU, Serbia—SRB, Slovenia—SVN, Turkey—TUR, Russia—RUS, Belarus—BLR, Lithuania—LTU, Latvia—LVA, Estonia—EST, Ukraine—UKR.

**Table 1 healthcare-11-00655-t001:** Values of the number of general practitioners per 100,000 inhabitants—value of the first observed year, last observed year, predictive value, median value, inter-incremental difference and linear regression analysis.

Countries	First Year (1990–2000)	Last Year (2014–2016)	Prediction	Median	IQR	Regression Analysis
Albania	52.9	55.86	60.63	53	3.94	y = −0.9346x + 56.908; R^2^ = 0.0911
Bulgaria	67.6	62.84	58.43	65	4.37	y = 2.7521x + 34.728; R^2^ = 0.283
Bosnia and Herzegovina	11.49	19.72	31.90	15	7.32	y = 1.4859x − 0.0295; R^2^ = 0.8872
Belarus	6.33	9.24	20.29	9	2.42	y = 0.6631x − 3.1129; R^2^ = 0.6046
Greece	14.31	39.15	52.88	18	6.91	y = 2.3669x − 5.6518; R^2^ = 0.8642
Croatia	55.02	57	62.33	53	4.65	y = 5.1864x − 20.139; R^2^ = 0.7321
Montenegro	30.48	39.18	46.01	33	7.68	y = 3.2036x − 3.3379; R^2^ = 0.746
Romania	65.82	56.95	51.34	65	9.16	y = 4.7354x − 12.363; R^2^ = 0.4253
Russia	38.7	32.09	49.38	43	14.62	y = 0.7576x + 37.779; R^2^ = 0.2111
Serbia	68.82	70.71	77.48	71	4.75	y = 4.8411x + 17.998; R^2^ = 0.5384
Slovenia	38.18	51.5	67.17	43	7.37	y = 4.338x − 5.5783; R^2^ = 0.8044
Turkey	48.26	53.47	64.72	51	8.19	y = 0.9193x + 43.067; R^2^ = 0.8606
Estonia	68.87	71.8	86.64	70	5.34	y = 1.8208x + 51.784; R^2^ = 0.6143
Lithuania	67.24	89.14	115.37	72	19.98	y = 2.4167x + 52.227; R^2^ = 0.95
Ukraine	31.78	36.11	46.84	33	7.12	y = 0.8446x + 25.892; R^2^ = 0.932

**Table 2 healthcare-11-00655-t002:** Values of the number of pharmacists per 100,000 inhabitants—value of the first observed year, last observed year, predictive value, median value, inter-incremental difference and linear regression analysis.

Countries	First Year (1990–1994)	Last Year (2014–2016)	Prediction	Median	IQR	Regression Analysis
Albania	38	84	72	39	6	y = 0.5234x + 21.161; R^2^ = 0.0276
Bulgaria	36	17	0	22	11	y = −1.5415x + 28.358; R^2^ = 0.7519
Bosnia and Herzegovina	18	12	11	10	1	y = 0.4002x + 2.293; R^2^ = 0.3079
Belarus	34	34	35	31	4	y = 0.049x + 30.474; R^2^ = 0.0163
Greece	86	105	129	96	13	y = 6.1128x − 32.852; R^2^ = 0.7909
Croatia	36	71	90	52	18	y = 1.6326x + 31.777; R^2^ = 0.985
North Macedonia	21	45	62	18	19	y = 0.7059x + 14.216; R^2^ = 0.1491
Montenegro	17	17	15	16	2	y = 0.9668x − 5.4213; R^2^ = 0.7147
Romania	29	73	114	46	31	y = 3.0312x − 7.7631; R^2^ = 0.6437
Russia	2	5	6	6	1	y = 0.1491x + 3.3187; R^2^ = 0.378
Serbia	25	33	42	30	6	y = 1.9579x − 9.5945; R^2^ = 0.8164
Slovenia	34	60	78	47	16	y = 2.99x − 4.5248; R^2^ = 0.8709
Turkey	29	35	38	34	2	y = 0.2371x + 30.43; R^2^ = 0.7305
Estonia	53	68	76	59	9	y = 1.5993x + 34.126; R^2^ = 0.4133
Latvia	56	78	100	63	12	y = 4.2935x − 21.52; R^2^ = 0.8292
Lithuania	52	66	87	59	3	y = −1.9902x + 49.605; R^2^ = 0.2204
Ukraine	3	3	5	3	1	y = 0.2082x − 0.6699; R^2^ = 0.8304

**Table 3 healthcare-11-00655-t003:** Values of the number of healthcare workers per 100,000 inhabitants—value of the first observed year, last observed year, predictive value, median value, inter-incremental difference and linear regression analysis.

Countries	First Year (1990–2000)	Last Year (2014–2016)	Prediction	Median	IQR	Regression Analysis
Albania	147	128	116	128	10	y = −3.0557x + 152.14; R^2^ = 0.2289
Bulgaria	298	400	426	353	24	y = 3.1217x + 316.19; R^2^ = 0.8513
Bosnia and Herzegovina	156	188	223	157	28	y = 6.9955x + 12.747; R^2^ = 0.3806
Belarus	288	407	446	327	45	y = 4.5933x + 276.18; R^2^ = 0.9225
Greece	363	625	815	466	219	y = 13.422x + 320.17; R^2^ = 0.9517
Croatia	194	313	357	241	41	y = 4.7335x + 186.48; R^2^ = 0.967
North Macedonia	234	280	315	232	41	y = −0.4363x + 235.11; R^2^ = 0.0034
Montenegro	193	219	241	204	13	y = 12.61x − 72.457; R^2^ = 0.7484
Romania	188	236	293	216	41	y = 10.824x − 0.7556; R^2^ = 0.5072
Russia	225	331	280	237	7	y = 3.7004x + 183.78; R^2^ = 0.2451
Serbia	275	307	355	300	28	y = 18.965x − 89.172; R^2^ = 0.7849
Slovenia	219	276	301	236	26	y = 13.403x − 0.2986; R^2^ = 0.7342
Turkey	97	175	218	140	49	y = 3.5474x + 94.603; R^2^ = 0.9923
Estonia	354	332	340	321	14	y = 0.1548x + 320.25; R^2^ = 0.012
Latvia	361	322	348	294	31	y = 0.8818x + 288.77; R^2^ = 0.0841
Lithuania	358	433	455	372	22	y = 6.1806x + 288.31; R^2^ = 0.2928
Ukraine	300	300	382	308	49	y = 19.561x − 45.216; R^2^ = 0.7598

**Table 4 healthcare-11-00655-t004:** Values of the number of dentists working per 100,000 inhabitants—value of the first observed year, last observed year, predictive value, median value, inter-incremental difference and linear regression analysis.

Countries	First Year (1990–1999)	Last Year (2016)	Prediction	Median	IQR	Regression Analysis
Albania	33.93	34.59	0	40	8.27	y = −2.5801x + 53.951 R^2^ = 0.9068
Bulgaria	67.95	100.38	109.97	82	19.01	y = 7.0214x + 71.396 R^2^ = 0.8403
Bosnia and Herzegovina	31.44	21.08	26.94	19	3.01	y = 1.8061x + 15.872 R^2^ = 0.7331
Belarus	31.72	54.89	69.26	44	15.53	y = 5.7146x + 36.692 R^2^ = 0.8768
Croatia	43.35	75.78	80.89	68	11.13	y = 2.9356x + 64.89 R^2^ = 0.8937
Montenegro	41.13	4.02	0	41	28.75	y = −24.34x + 80.441 R^2^ = 0.8498
Romania	31.65	67	98.34	49	22.15	y = 14.54x + 18.098 R^2^ = 0.9328
Russia	26.95	29.22	28.58	29	1.94	y = −0.2971x + 30.108 R^2^ = 0.7027
Slovenia	59.06	64.93	68.43	60	2.10	y = 2.5453x + 54.943 R^2^ = 0.9631
Estonia	51.75	89.68	100.75	79	22.56	y = 5.2017x + 73.115 R^2^ = 0.8348
Latvia	55.24	90.54	115.01	67	14.43	y = 11.234x + 51.215 R^2^ = 0.8855
Ukraine	45.43	68.37	100.43	46	20.32	y = 13.211x + 25.66 R^2^ = 0.8721

**Table 5 healthcare-11-00655-t005:** Indicator values of total number of scanners for computed tomography per 100,000 inhabitants—value of first observed year, last observed year, predictive value, median value, inter-incremental difference and linear regression analysis.

Countries	First Year (2005)	Last Year (2016)	Prediction	Median	IQR	Regression Analysis
Bulgaria	1.6	3.5	5.2	3.0	1.4	y = 0.1892x + 1.5121 R^2^ = 0.9188
Estonia	0.7	1.8	2.6	1.6	0.6	y = 0.1049x + 0.8015 R^2^ = 0.7725
Greece	2.5	3.7	4.5	3.3	0.5	y = 0.1x + 2.5333 R^2^ = 0.9429
Croatia	1.6	1.8	1.8	1.6	0.2	y = 0.0129x + 1.48 R^2^ = 0.0643
Lithuania	1.2	2.3	3.5	1.9	1.1	y = 0.1294x + 0.9424 R^2^ = 0.8095
Latvia	1.8	3.6	5.5	3.0	1.4	y = 0.1962x + 1.55 R^2^ = 0.9615
Romania	0.3	1.3	2.4	0.8	0.7	y = 0.1152x − 0.0636 R^2^ = 0.9945
Slovenia	1	1	0.8	1.1	0.2	y = −0.0112x + 1.1727 R^2^ = 0.1119
Serbia	1.3	1.4	1.6	1.3	0.1	y = 0.03x + 1 R^2^ = 0.45
Turkey	0.7	1.5	2.0	1.3	0.4	y = 0.0671x + 0.7636 R^2^ = 0.8951

**Table 6 healthcare-11-00655-t006:** Values of total number of magnetic resonance units per 100,000 inhabitants—value of the first observed year, last observed year, predictive value, median value, inter-incremental difference and linear regression analysis.

Countries	First Year (2005)	Last Year (2016)	Prediction	Median	IQR	Regression Analysis
Bulgaria	0.3	0.8	1.3	0.5	0.4	y = 0.0524x + 0.1758 R^2^ = 0.9008
Estonia	0.2	1.4	2.1	0.9	0.5	y = 0.0941x + 0.247 R^2^ = 0.9377
Greece	1.3	2.7	3.4	2.2	0.5	y = 0.0976x + 1.4742 R^2^ = 0.8353
Croatia	0.3	0.4	0.5	0.3	0.1	y = 0.0123x + 0.2234 R^2^ = 0.5033
Lithuania	0.2	1.2	2.2	0.6	0.8	y = 0.101x + 0.0348 R^2^ = 0.9305
Latvia	0.3	1.4	2.3	0.9	0.7	y = 0.1021x + 0.1864 R^2^ = 0.9743
Romania	0.1	0.6	1.1	0.4	0.3	y = 0.0576x − 0.1018 R^2^ = 0.9733
Slovenia	0.6	1.1	1.4	0.8	0.2	y = 0.0445x + 0.4882 R^2^ = 0.9095
Serbia	0.3	0.3	0.3	0.3	0	/
Turkey	0.3	1.1	1.6	1.0	0.4	y = 0.0643x + 0.4152 R^2^ = 0.7927

## Data Availability

The data sets used and/or analyzed in the present study are available on https://gateway.euro.who.int/en/datasets/european-health-for-all-database/ (accessed on 21 November 2022).

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
