# Peer review of "Analysis and Forecast of Indicators Related to Medical Workers and Medical Technology in Selected Countries of Eastern Europe and Balkan"

_healthcare, 2023, doi:10.3390/healthcare11050655_

Round 1

Reviewer 1 Report

Dear authors,

Congratulations on your work. I believe that the manuscript would be improved if you address the following topics:

- Page 2, line 63: the consequence is the rise of chronic diseases only? I believe that other types of diseases/disorders could rise from that...

- Page 2, line 76: replace counties with countries

- Page 2, line 77: "... with similar historical background in development of health systems". What do you mean by similar? What kind of parameters is similar? You should explore/explain

- Page 2, lines 86-87: why do Pharmacists and General practitioners are designed as "(PP)"? 

- If HFA-DB provides information for 53 countries, what is the interest in the selected countries (Albania, Bulgaria, Bosnia and Herzegovina, Belarus, 91 Greece, Croatia, North Macedonia, Montenegro, Romania, the Russian Federation, Ser- 92 bia, Slovenia, Turkey, Estonia, Lithuania, Latvia, and Ukraine)? This selection must be better explained.

- Could you provide scientific references for the "linear trend estimate"  used?

- "interquartile 100 range 25–75th percentile were used for better comparison of each country". How the 25-75th percentile allow a better comparison? And also, what do you mean with better?

- "Quantitative forecasting technique". Once again, you should provide one or two scientific references in this paragraph

- Tables 1 to 6: When you refer "first year" and "last year" you should add the year between ().

- In the results/discussion section you identify a lot of parameters and trends for the countries selected. In the discussion, I am expecting more development... e.g: what do you conclude from the trends? Any "allert" situation/trend? Do you suggest any measure or future study?

Thank you.

Author Response

Dear reviewer,

Thank you for valuable comments and with your directions, I believe that our work has improved a lot.

I will try my best to respond on your suggestion.

  1. Page 2, line 63: the consequence is the rise of chronic diseases only? I believe that other types of diseases/disorders could rise from that...

Answer: changed to: communicable and non-communicable diseases (this way we covered wider range of diseases)

  1. Page 2, line 76: replace counties with countries

Answer: changed

  1. Page 2, line 77: "... with similar historical background in development of health systems". What do you mean by similar? What kind of parameters is similar? You should explore/explain

Answer: Eastern Europe and Balkan, Russian Federation and former Unite of Soviet Social Republics and Turkey, are countries that shared historical background so as the way of how they managed different economic crises that affect their similar health system organization. These countries are that have very diverse structure of population when looking the religion (Catholicism, orthodox Christianity and Islam), which is very important because of the way how people historically had an approach towards making important solutions for the countries, when comparing to the other parts of Europe – northern and southern. Developing the health systems, these countries also had very dependent relationship between each other as the health systems that these countries used where the ones that other adapted to their countries. There were 3 health systems of health financing that were dominantly throughout 19th century - the Bismarck, Beveridge, and Semashko systems. Progress in medical technology and pharmaceuticals were hard to follow after the other, more developed countries of the Europe and countries that we selected have large problems with accessing the medical health care especially in the rural and less developed parts of countries. A lot of medical expenditures were played out-from-pocket. Those countries were less industrial developed at the time, especially during the Cold war, and lot of GDP came from their agricultural economy which couldn’t endure such a fast medical development, especially later when the population aging came from the shadow. Socioeconomic situation of people from these countries was weak so the health care was way expensive and less availably to the all. (imported into the Introduction)

  1. Page 2, lines 86-87: why do Pharmacists and General practitioners are designed as "(PP)"? 

Answer: PP is aberration for practicing physician/persons. All indicators that we used in this article are ones that are providing any kind of medical health, not ones that are in educational process. Every used indicator has an inclusion and exclusion criteria that are defined by the World Health Organization before inputted in the HFA-DA (Imported into the Methodology). example: https://gateway.euro.who.int/en/indicators/hfa_508-5291-number-of-general-practitioners-pp/

  1. If HFA-DB provides information for 53 countries, what is the interest in the selected countries (Albania, Bulgaria, Bosnia and Herzegovina, Belarus, 91 Greece, Croatia, North Macedonia, Montenegro, Romania, the Russian Federation, Ser- 92 bia, Slovenia, Turkey, Estonia, Lithuania, Latvia, and Ukraine)? This selection must be better explained.

Answer: It is explained in the paragraph about historical background and development of health system in these countries. References are added.

  1. Could you provide scientific references for the "linear trend estimate" used?

Answer: It is now provided and imported into the methodology as the reference.

  1. "interquartile 100 range 25–75th percentile were used for better comparison of each country". How the 25-75th percentile allow a better comparison? And also, what do you mean with better?

Answer: – This sentence is now rewritten. It was meant to be written easier (this adjective better is replaced), but it was meant mainly on the median value as it gives the single digit easier to compare with other values. The porpoise of IQR was explained in the Methodology.

  1. "Quantitative forecasting technique". Once again, you should provide one or two scientific references in this paragraph

Answer: Forecasting was done by regression analysis. References are provided into the Methodology section.

  1. Tables 1 to 6: When you refer "first year" and "last year" you should add the year between ().

Answer: It is changed according to this comment, the range of years I added in the tables.  

  1. In the results/discussion section you identify a lot of parameters and trends for the countries selected. In the discussion, I am expecting more development... e.g: what do you conclude from the trends? Any "allert" situation/trend? Do you suggest any measure or future study?

Answer: Opinion about our results is written in the Conclusion part. Recommendations about new studies are added at the end of conclusion paragraph.

We would recommend more studies like this in the future, so that we can follow if changes continues to be positive, also the assessment of the medical workers and technologies can help navigate the best resource allocation for medical healthcare. Assessing the trends in more developing countries can help the less developing countries to emulate to their healthcare organization and improve.

For more details please see the revised manuscript.

Reviewer 2 Report

1. The data source used in the paper is outdated. Could you find the more recent dataset for analysis?

2. All the table labels are the "First Year" and "Last Year", is it for "The year 1990" and "The year 2014" respectively?

3. Please consider more factors affecting such as the environmental changes and healthcare service requirements changes when you predicate long periods like 2 to 3 decades of development needed.

Author Response

Dear reviewer,

Thank you for valuable comments and with your directions, I believe that our work has improved a lot.

I will try my best to respond on your suggestion.

  1. The data source used in the paper is outdated. Could you find the more recent dataset for analysis?

Answer: The 2014-2016 are the last year with available data for observed indicators, so these values are currently the newest in the database. Last updated of Health for All (HFA-DB) was at 01 September 2022 and values from this research are using those data. This been now stated in the Methodology. Since the mid-1980s, Member States of the WHO European Region have been reporting essential health-related statistics to the Health for All (HFA) family of databases, making it one of WHO’s the oldest sources of data. As it is based on reported data, rather than estimates, the HFA family of databases is also particularly valuable.

  1. All the table labels are the "First Year" and "Last Year", is it for "The year 1990" and "The year 2014" respectively?

Answer: We have labeled years as the first year and last year, because the first year of followed indicator differed from country to country. For the most countries the most similar year with available value was used, and it was between 1990 to 1999 deepening on the indicator. Same for the last year, the last year from HFA-DB was 2014 or 2016. This explanation is imported in the Methodology. This was changed in the tables and expressed as the range of years.

  1. Please consider more factors affecting such as the environmental changes and healthcare service requirements changes when you predicate long periods like 2 to 3 decades of development needed.

Answer: It is imported throughout text. We did smaller prediction, only one decade, so we can minimize the possibility of errors. These predictions help us to focus towards the issues that negative change can bring and prevent it as well as possible. Global aging and re-composition of people from rural to urban areas are something that is real and happening. The number of people age 65+ is in continues rise, as like the leaving from less to more developed part of countries or into the different countries. This migration, especially of medical workers will influence the healthcare accessibility of people in need, especially to older people.

For more details please see the revised manuscript.

Reviewer 3 Report

The study used linear regression analysis to access the timeline changes in observed indicators for each country and to calculate the progress of these countries over time. There are several questions needed be elucidated by the authors before further review.

I wonder if the analysis is appropriated. How the authors can assume the trend is linear? What are the parameters and sample the study used for a time-series analysis? It can be hard to estimate a time trend with only 25 yearly point. The model fits are various cross the countries and indicators, in which the values of R square are from 0.01 to 0.99. What is the confidence interval for the prediction by 2025?  

The assessment of health service system includes not only the number but allocation of health resources. Demographic structure of population, labor regulation (e.g., work hour) and health policy (e.g., insurance, disease prevention program) may affect the necessity and accessibility of people in need as well. The study barely comparing the difference of country-level numbers over time cross the countries may have neglected the complexity of health system.

As I mention above, lack of the theory of study design makes the work hard to achieve the study purpose and contribute to acknowledge.

Author Response

Dear reviewer,

Thank you for valuable comments and with your directions, I believe that our work has improved a lot.

I will try my best to respond on your suggestion.

  1. The study used linear regression analysis to access the timeline changes in observed indicators for each country and to calculate the progress of these countries over time. There are several questions needed be elucidated by the authors before further review. I wonder if the analysis is appropriated. How the authors can assume the trend is linear? What are the parameters and sample the study used for a time-series analysis? It can be hard to estimate a time trend with only 25 yearly point. The model fits are various cross the countries and indicators, in which the values of R square are from 0.01 to 0.99. What is the confidence interval for the prediction by 2025?

Answer: It is a descriptive data analysis of selected health indicators extracted from HFA-DB, they are fully available and tracked by the World Health Organization by large number of countries since 1980s. The data provided from HFA datasets are time (years) and values of observed indicators throughout that time. Countries without consistent follow of certain indicator were not included in the analysis. As we are observing changes of these indicators only through time (continuous variable), linear trend is appropriate analysis. Years that are observed varies from the countries so for the first year it was taken the value from the year that most of the countries had in common (this way we excluded possibilities for large differences between first year of observation which would influence the comparison; for the most countries that was year 1990 –but it varied and range of first and list observed year are added in the tables) and for the last year is used the year 2014 or 2016 (the last available year after the update of HFA-DB, in September 2022). With this data we were able to access current simple linear trend using the excel mathematics algorithm, and we got the graphs that descriptively showed us the changes in those trends. Linear regression was performed using the SPSS, and using it we predicted values based on the data from available two and half decades, only one decade after the last available year was predicted, with purpose of seeing will the trends continue to follow current ones. Larger predictions could result with chances of error so using the smaller prediction values gave more accurate results. Different values of R square are different for different indicators due to not identical periods available when observed. Confidence interval for prediction was 95%.

  1. The assessment of health service system includes not only the number but allocation of health resources. Demographic structure of population, labor regulation (e.g., work hour) and health policy (e.g., insurance, disease prevention program) may affect the necessity and accessibility of people in need as well. The study barely comparing the difference of country-level numbers over time cross the countries may have neglected the complexity of health system.

Answer: We agree with those facts. Health service system is depending on multiple factors, and those factors are mentioned thought the introduction and discussion part (it is now updated according to your comment). There is no standardized indicator for work hour, insurance, diseases prevention program nor were they followed for many decades, so we couldn’t analyze those indicators but they are mentioned as the reasons that influences the health system. This is a descriptive analysis with purpose to show the current status about the indicators with known values, indicators that we used are standardized and followed for many decades by countries that report their values to the WHO. Indicator of demographic structure is available (for the population aged 65+) and we have analyzed it but not included in the results of this research. It showed us the increasing trend in this age group, which is in the line with current global aging trends that is connected to the necessary increasing number of medical workers and technologies (it is mentioned in the manuscript). Medical workers and medical technologies are the key of health system and their numbers must follow the necessity of population. The purpose of this article is to reevaluate current trends of those indicators in the countries from Eastern Europe and Balkan – countries with same historical background and similar development of health systems with reference to their development level and multi-culture.

For more details please see the revised manuscript.

Reviewer 4 Report

A descriptive data analysis of selected health indicators, extracted from 53 countries and 153 health indicators was studied. The results can help government and health sector to focus and navigate the best investments of each country according to the level of their development. This study is very interesting and useful. My comments are:

1. Check all the words and logic.

2. The Statistical methods and software should be given.

3. The conclusion should be more clear and concise.

Author Response

Dear reviewer,

Thank you for valuable comments and with your directions, I believe that our work has improved a lot.

I will try my best to respond on your suggestion.

  1. Check all the words and logic.

Answer: All words are checked and some of sentences were re-written.

  1. The Statistical methods and software should be given.

Answer: More description about methods and software are added.

  1. The conclusion should be more clear and concise.

Answer: Conclusion is rewritten.

For more details please see the revised manuscript.

Round 2

Reviewer 1 Report

Dear authors,

I appreciate your answer. I think your manuscript has been clearly improved.

Author Response

Dear authors,

I appreciate your answer. I think your manuscript has been clearly improved.

Answer:

Dear Reviewer,

Thank you for your contribution in improving of our work, we really appreciate your suggestions.

Reviewer 2 Report

Yes. Thank you for your time in modification of your paper. It looks better this time. I appreciate that you spared more time to re-consider and re-structure this paper.

You added "We would recommend more similar studies in the future, with purpose of following the change; also the assessment of the medical workers and technologies can help navigate the best resource allocation for medical healthcare. Assessing the trends in more developing countries can help other countries to emulate to their healthcare organization and improve." in your Conclusions. Would you please explain the aim of it?

Actually, these kinds of studies never stopped no matter in developed countries or developing countries. All their governments are considering the balance of the costs of healthcare and population health development. How to take care of their population's health belongs to another topic that needs to be separately discussed.

Author Response

Dear reviewer,

Thank you on the time for the second review of our work. I have answer on your comments.

  1. You added "We would recommend more similar studies in the future, with purpose of following the change; also the assessment of the medical workers and technologies can help navigate the best resource allocation for medical healthcare. Assessing the trends in more developing countries can help other countries to emulate to their healthcare organization and improve." in your Conclusions. Would you please explain the aim of it?

Actually, these kinds of studies never stopped no matter in developed countries or developing countries. All their governments are considering the balance of the costs of healthcare and population health development. How to take care of their population's health belongs to another topic that needs to be separately discussed.

Answer: I see the point of this comment and I’ll try to explain our context better, and modified the paragraph in the Conclusion.

Importance of observing different health and economic related indicator is useful on the many different country levels and their analysis can be reflection of successful application of preventing measures. Assessing these trends and comparison between countries may give valuable information about organization of different health systems and countries with specific problem can adapt their health system to it, according to their possibilities.

Reviewer 3 Report

The authors have responded to part of my questions. However, the study’s serious flaws in analysis make the research contribute less valuable information. I raise two major concerns below.

1.The authors state the study is a descriptive data analysis. Why was a linear regression performed? Simple linear regression is used to discriminate (not just describe) relationship between two variables, which can be established with a significant statistics value only if the assumption of normal distribution is satisfied and no other factor interfere. I don’t think it can be used appropriately to analyze the time trend and predict the timely change of indicators. If the study’s finding is presented with a descriptive data analysis only and can’t be proved by a significant testing, what will the authors like to highlight for readers?

2.The authors state those indicators are standardized before analyses. What kind of standardization did they use? They address the populations had been getting older over time. Is there any treatment using age-standardization in the analysis? If no, how can the indicators be compared cross the countries or years?

Author Response

Dear reviewer,

Thank you for your answer.

I hope that our comments would help you with your concerning parts and you can understand the methodology and aim of our work.

  1. The authors state the study is a descriptive data analysis. Why was a linear regression performed? Simple linear regression is used to discriminate (not just describe) relationship between two variables, which can be established with a significant statistics value only if the assumption of normal distribution is satisfied and no other factor interfere. I don’t think it can be used appropriately to analyze the time trend and predict the timely change of indicators. If the study’s finding is presented with a descriptive data analysis only and can’t be proved by a significant testing, what will the authors like to highlight for readers?

Answer: The study is defined as descriptive, because part of the study is based on the comparison of the descriptive data (measure of central tendency – median, measure of dispersion or variation - interquartile range) that we got from statistical methods. Median value that we got from the SPSS analysis is used for comparison between countries for each indicator, and graphics are also made with that purpose. We can remove that adjective if it is confusing. There are no other factors to interfere in our analysis, it was just years and values of selected indicators that we analyzed using linear regression that was used in the similar researches as well; we added few references with researches that used regression in their results. Normality is proven with Shapiro test as the sample size is less than 50. Reports of this kind are done in many big analyses for the national reports. This is a study that points the progress in these indicators for multiple countries (18 countries) so that we can compare the progress of countries with historical similar pattern of development of health system, as is stated in the introduction and aims of the study.

  1. The authors state those indicators are standardized before analyses. What kind of standardization did they use? They address the populations had been getting older over time. Is there any treatment using age-standardization in the analysis? If no, how can the indicators be compared cross the countries or years?

Answer: As stated in the article and previous answer, the indicators we have used are from HFA-DB, these reports are submitted to the WHO and updated when new data is received. We did not perform standardization of any kind nor our research used or compared indicators related to the certain age group; we have collected the data sets and performed described analysis. These reports (values) are received from the different countries that are using the appropriate techniques in collecting these indicators for many years; they have also described all indicators in details with statement of the inclusion and exclusion criteria (example: https://gateway.euro.who.int/en/indicators/hfa_508-5291-number-of-general-practitioners-pp/). They standardized these indicators for usage so the progress of many different indicators can be followed through time and with that knowledge the appropriate measures can be taken on the different levels (national, local etc.), depending of the indicator of interest. In our case, we did research of indicators of medical workers and medical technologies with a narrative review of the factors that influences the change and explained its importance.